# The effect of univariate bias adjustment on multivariate hazard estimates

**Jakob Zscheischler**[1,2,3]**, Erich M. Fischer**[1]**, and Stefan Lange**[4]

[1]Institute for Atmospheric and Climate Science, ETH Zurich, Universitaetstrasse 16, 8092 Zurich, Switzerland
[2]Climate and Environmental Physics, University of Bern, Sidlerstrasse 5, 3012 Bern, Switzerland
[3]Oeschger Centre for Climate Change Research, University of Bern, 3012 Bern, Switzerland
[4]Potsdam Institute for Climate Impact Research (PIK), Member of the Leibniz Association, P.O. Box 60 12 03, 14412 Potsdam, Germany

**Correspondence:** Jakob Zscheischler (jakob.zscheischler@climate.unibe.ch)

**Abstract.** Bias adjustment is often a necessity in estimating climate impacts because impact models usually rely on unbiased climate information, a requirement that climate model output rarely fulfills. Most currently used statistical bias adjustment methods adjust each climate variable separately, even though impacts usually depend on multiple, potentially dependent variables. Human heat stress, for instance, depends on temperature and relative humidity, two variables that are often strongly correlated. Whether univariate bias adjustment methods effectively improve estimates of impacts that depend on multiple drivers is largely unknown, and the lack of long-term impact data prevents a direct comparison between model output and observations for many climate related impacts. Here we use two hazard indicators, heat stress and a simple fire risk indicator, as proxies for more sophisticated impact models. We show that univariate bias adjustment methods such as univariate quantile mapping often cannot effectively reduce biases in multivariate hazard estimates. In some cases, it even increases biases. These cases typically occur (i) when hazards depend equally strongly on more than one climatic driver, (ii) when models exhibit biases in the dependence structure of drivers, and (iii) when univariate biases are relatively small. Using a perfect model approach, we further quantify the uncertainty of bias-adjusted hazard indicators due to internal variability and show how imperfect bias adjustment can amplify this uncertainty. Both issues can be addressed successfully with a statistical bias adjustment that corrects the multivariate dependence structure in addition to the marginal distributions of the climate drivers. Our results suggest that currently many modelled climate impacts are associated with uncertainties related to the choice of bias adjustment. We conclude that in cases where impacts depend on multiple dependent climate variables these uncertainties can be reduced using statistical bias adjustment approaches that correct the variables' multivariate dependence structure.

## 1 Introduction

With ongoing climate change, climate impact modelling has become an important pillar of climate research, informing decision makers and risk managers in many sectors that are affected by climate variability. Impact models such as hydrological models, crop models and epidemiological models usually rely on absolute thresholds in their driving climate variables such as temperature, precipitation, wind speed and humidity (Winsemius et al., 2015; Ruane et al., 2017) and thus require unbiased climate information as input. However, biases continue to persist in global (Flato et al., 2013; Wang et al., 2014) and regional climate models (Christensen et al., 2008; Kotlarski et al., 2014), rendering bias adjustment an undesired but often unavoidable data processing step for climate impact modeling (Piani et al., 2010).

The need for easily available information on climate impacts has led to a sometimes overly uncritical use of bias adjustment in large impact modeling projects, even though its current usage might in some cases result in ill-informed adaptation decisions (Maraun et al., 2017). It is often assumed that biases are stationary, hence that bias adjustment

can be developed in current climate and applied in a warmer world. It is not clear, however, for which variable and to what extent this assumption is met. Moreover, while uncertainties related to the choice of climate model and impact model as well as future socio-economic scenarios are frequently carried through the modeling chain (Wilby and Dessai, 2010; Frieler et al., 2017), uncertainties associated with the choice of the applied bias adjustment method are rarely reported (Ehret et al., 2012) though there are exceptions (Chen et al., 2011; Bosshard et al., 2013; Addor and Fischer, 2015).

Assessing the appropriateness of bias adjustment for impact modeling is not an easy task (Papadimitriou et al., 2017). While the quality of bias adjustment on climate variables can be evaluated against observed quantities in the climate domain, the implications for modeled impacts are much harder to determine. On the one hand, limited length and a lack of homogeneity in observed climate impacts render an evaluation of modeled impacts very challenging (Cramer et al., 2014). On the other hand, many impacts rely on the complex interaction of multiple climate variables across time and space (Zscheischler et al., 2018), preventing an evaluation in the climate domain. For instance, temperature and precipitation variability affect crop yields (Semenov and Porter, 1995; Zscheischler et al., 2017) and other ecosystem services such as net carbon uptake (Humphrey et al., 2018). Flood occurrence, its intensity and associated damages depend on the temporal and spatial characteristics of precipitation, soil moisture, river flow and surge (Vorogushyn et al., 2018). Drought-related impacts depend on precipitation, evapotranspiration and temperature, their spatial distribution, and their interaction with human activities (Van Loon et al., 2016). Fire occurrence and strength not only requires available fuel and an ignition source, but is also strongly dependent on relative humidity and temperature (Brando et al., 2014). Infrastructural damage is particularly large when strong winds and extreme precipitation occur jointly (Martius et al., 2016). Finally, climate-related human health impacts are linked to a suite of climatic drivers (McMichael et al., 2006). It is currently largely unknown how widely used statistical bias adjustment approaches affect modeled impacts that depend on multiple drivers such as the ones mentioned above.

Bias adjustment methods are often designed to correct one variable at a time (Teutschbein and Seibert, 2012; Hempel et al., 2013). A possible bias in the dependence structure between climate variables is therefore not adjusted, even though climate models may not capture dependencies between climate drivers very well, for instance the correlation between temperature and precipitation in summer (Zscheischler and Seneviratne, 2017). Even worse, correcting a bias in marginal distributions may modify the dependence between variables. In recognition of these issues, several multivariate bias adjustment methods have been suggested in the recent past (Piani and Haerter, 2012; Li et al., 2014; Vrac and Friederichs, 2015; Cannon, 2016; Mehrotra and Sharma, 2016; Vrac, 2018). In addition to adjusting the marginal distributions, these methods adjust, to a certain extent, the dependence structure between multiple variables. Some studies have recently suggested that multivariate approaches do not lead to a substantial improvement for certain specific regional impacts (Yang et al., 2015; Casanueva et al., 2018; Räty et al., 2018). However, this does not imply that multivariate bias adjustment is not necessary in any case. Many multivariate methods have not been evaluated systematically against modeled impacts and have been rarely used in larger-scale impact modeling frameworks, though there are notable exceptions. For instance, Cannon (2018) presents a highly flexible multivariate bias adjustment method and demonstrates its effectiveness on a modeled multivariate hazard indicator. By adjusting biases and dependencies between temperature, precipitation, relative humidity and wind speed, the method substantially reduces biases in the five-dimensional Fire Weather Index (FWI), outperforming univariate bias adjustment approaches.

In this paper we study the limitations of univariate bias adjustment methods for multivariate impacts. We focus on global-scale climate output from global circulation models (GCMs), which is frequently used for global impact studies (Winsemius et al., 2015; Frieler et al., 2017; Ruane et al., 2017). However, our analyses are of a rather conceptual nature and thus also apply to other climate model output, for instance from regional climate models. First, we investigate whether traditional univariate bias adjustment methods generally reduce biases in impacts that depend on multiple dependent drivers. Bias adjustment may also amplify uncertainties inherent to the observations (Chen et al., 2011). Hence, using a model environment with multiple models and multiple runs for a single model, we further estimate how uncertainties related to internal variability may be propagated and amplified through incomplete bias adjustment. We use multivariate hazard indicators as proxies for actual impacts to evaluate the effect of bias adjustment on impacts and to overcome the challenge of missing impact data. It can be assumed that "real" impacts are in many cases more complex and also depend on more driving variables. Therefore, our analysis provides a rather conservative estimate on the potential effects of univariate bias adjustment on modeled impacts.

## 2  Data and Methods

### 2.1  Data

*Observations.* We use daily temperature and relative humidity from the ERA-Interim reanalysis (Dee et al., 2011) as main reference data set. We further use daily temperature and relative humidity from the observational data set EWEMBI (Lange, 2016, 2018), which has been used in the Inter-Sectoral Impact Model Intercomparison Project phase 2b (ISIMIP2b Frieler et al. 2017).

*Model simulations.* We use daily model output from the historical runs of the Coupled Model Intercomparison

Project phase 5 (CMIP5, Taylor et al. 2012). A main objective of the CMIP5 model experiments is to improve our understanding of the climate system, and to provide estimates of future climate change. All 29 model simulations used in this study are listed in Table A1.

All data have been bilinearly interpolated to a 2.5 by 2.5 degree regular latitude-longitude grid prior to analysis. The studied time period is 1981-1995. Our analysis is based on daily values during the hottest month of the year at each grid point, which results in about 450 samples. We focus on the hottest month to avoid dealing with seasonality and because, arguably, fire risk and heat stress are most relevant during this time period. The hottest month was identified based on the climatology of ERA-Interim monthly temperature data. Although the effectiveness of bias adjustment is typically evaluated outside the calibration period (Maraun, 2013), here we focus our analysis completely on the selected 15-year period. This allows a separate assessment of the effect of bias adjustment on univariate versus multivariate impacts independently from other effects such as cross validation error. A good performance in the calibration period is a necessary requirement for an effective bias adjustment approach. Furthermore, cross validation might not help to diagnose whether a bias adjustment approach is effective (Switanek et al., 2017; Maraun and Widmann, 2018).

## 2.2 Methods

*Hazard indicators.* We use the Wet Bulb Globe Temperature (WBGT, Dunne et al. 2013) as an indicator for heat stress and the Chandler Burning Index (CBI, Chandler et al. 1983) as an indicator for fire risk. Both are relatively simple indicators that can be computed solely from daily temperature and relative humidity. WBGT and its variants have been used extensively to assess projections of heat stress under climate change (Pal and Eltahir, 2015; Zhao et al., 2015; Li et al., 2017). CBI is one out of many fire risk indicators (Lee, 1980; Carlson and Burgan, 2003), here mainly chosen for its simplicity. The hazard intensities of WBGT and CBI vary along different gradients in the temperature-humidity domain (Fischer et al., 2013; Zscheischler et al., 2018). WBGT increases with hotter and more humid conditions and is equally dependent on temperature and humidity (Fig. 1). In contrast, CBI increases with hotter and drier conditions and its variability is mostly driven by humidity and much less by temperature. Using two hazard indicators that depend differently on the same climatic drivers enables us to study how the relationship between hazard direction and driver distribution changes the way bias adjustment affects modeled hazards (Zscheischler et al., 2018). While certainly more sophisticated indicators exist, both for fire risk and heat stress (Bröde et al., 2013), our goal here is rather to provide an illustrative example of the issues associated with bias adjustment and impact modeling than to provide the most reliable hazard projections. We estimate WBGT following the approach outlined

in the Supplementary Material of Dunne et al. (2013). CBI can be computed as

$$CBI = \left(\left(\left(110 - 1.373RH\right) - 0.54(10.2 - T)\right)124 \cdot 10^{-0.0142RH}\right)/60 \tag{1}$$

where $RH$ is relative humidity in $\%$ and $T$ is temperature in degree Celsius.

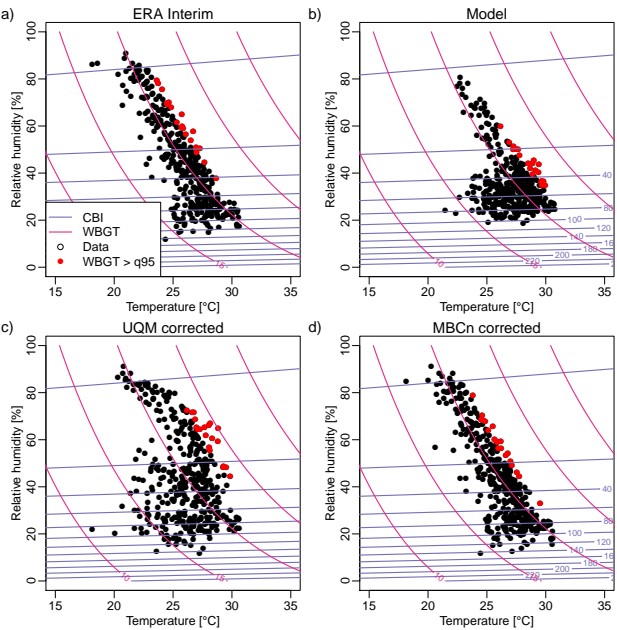

**Figure 1.** Distribution of daily temperature versus relative humidity during July (1981-1995) for a grid point in central Africa. ERA-Interim (a), model simulations from the model IPSL-CM5A-MR (b), bias-adjusted with UQM (c), bias-adjusted with MBCn (d). Pink lines depict levels of equal heat stress (WBGT). Violet lines depict levels of equal fire risk (CBI). Red points denote values for which WBGT exceeds its 95th percentile.

*Bias adjustment.* We employ two different bias adjustment methods, the widely used univariate empirical quantile mapping (UQM, Panofsky and Brier 1968; Maraun 2013; Casanueva et al. 2018) and the multivariate bias adjustment in $n$ dimensions (MBCn) developed by Cannon (2018). UQM applies separate corrections to a fixed number of quantiles to adjust a modeled empirical cumulative distribution towards the observed empirical cumulative distribution. Hence, if $X_o$ and $X_m$ be the observed and modelled values, respectively, then

$$\hat{X}_m = F_o^{-1}(F_m(X_m)) \tag{2}$$

where $F_m$ is the empirical cumulative distribution function of $X_m$ and $F_o^{-1}$ is the inverse empirical distribution function (or quantile function) corresponding to $X_o$. Values in between the pre-defined quantiles are approximated using linear interpolation. We apply UQM with the R package qmap (Gudmundsson, 2014) using 100 quantiles.

MBCn is a bias adjustment method in which both the marginal distribution of each individual variable and the multivariate dependence structure are corrected at the same time (Cannon, 2018). This is achieved by an iterative approach, which first applies a random rotation $R^{[j]}$ to the multivariate observed and modeled data distribution

$$\tilde{X}_m^{[j]} = X_m^{[j]} R^{[j]} \tag{3}$$
$$\tilde{X}_o^{[j]} = X_o^{[j]} R^{[j]}. \tag{4}$$

Subsequently, quantile mapping (Equation 2) is applied to the rotated data $\tilde{X}_m^{[j]}$ with $\tilde{X}_o^{[j]}$ as a reference, yielding $\hat{X}_m^{[j]}$. Then the inverse rotation is applied

$$X_m^{[j+1]} = \hat{X}_m^{[j]} R^{[j]-1}. \tag{5}$$

The observed data is carried forward to the next iteration unchanged $X_o^{[j+1]} = X_o^{[j]}$. These steps are repeated until the modeled data distribution has converged to the observed distribution. One may interpret the algorithm such that the random rotations allow an information exchange between the different dimensions.

Finally, we use the bias adjustment method (Lange, 2017) applied in ISIMIP2b (Frieler et al., 2017). This method adjusts relative humidity with parametric quantile mapping using beta distributions to model simulated and observed daily values (Frieler et al., 2017; Lange, 2018), and temperature with an additive correction of monthly mean temperature and a multiplicative correction of daily anomalies from the monthly mean temperature (Hempel et al., 2013).

We adjust daily values of temperature and relative humidity in CMIP5 during the hottest month at each grid point and evaluate the change in bias for WBGT and CBI.

*Perfect model approach.* Internal variability can lead to uncertainties in the bias adjustment, which may be amplified through an inadequate choice of bias adjustment. Since fully-coupled ocean-atmosphere models provide different realizations of unforced internal variability, observations and models as well as different simulations of the same model are not expected to agree on year-by-year basis. Even when using a time period of 2-3 decades this constitutes a substantial source of uncertainty (Addor and Fischer, 2015). Using the multimodel environment of CMIP5, we study whether UQM increases uncertainties related to internal variability for multivariate hazard indicators. We conduct a perfect model approach to separate uncertainties associated with internal variability and the choice of bias adjustment (Griffies and Bryan, 1997; Elía et al., 2002; Hawkins et al., 2011). To this end we use the five available initial condition members of the model CanESM to estimate the influence of internal variability as a source of uncertainty. First, we bias adjust CanESM runs against each other to estimate the range of the uncertainty due to internal variability ("noise"). Bias adjusting all other model runs against all five CanESM runs then provides an estimate for the full uncertainty range ("full range"). Comparing the range of the "noise" with the "full range", we estimate whether univariate bias adjustment amplifies uncertainties related to internal variability.

## 3   Results

Statistical bias adjustment that is applied separately on each marginal of a multivariate distribution such as UQM ensures that the bias adjusted modeled distributions of the marginals are well aligned with the observed marginal distributions. We illustrate the effect of bias adjustment for one selected grid point in central Africa. UQM "squeezes" and "stretches" a modeled multivariate distribution along the marginal axes to match the observations of the marginals (Fig. 1c). For hazards that are a function of multiple drivers and that vary along a diagonal gradient such as WBGT, UQM may not be able to reduce biases. In fact, for many percentiles, it can even increase biases as illustrated in Fig. 2c (blue dots). CBI seems to be less affected, likely because its variability is mostly driven by a single variable, namely relative humidity (Figs. 1-2).

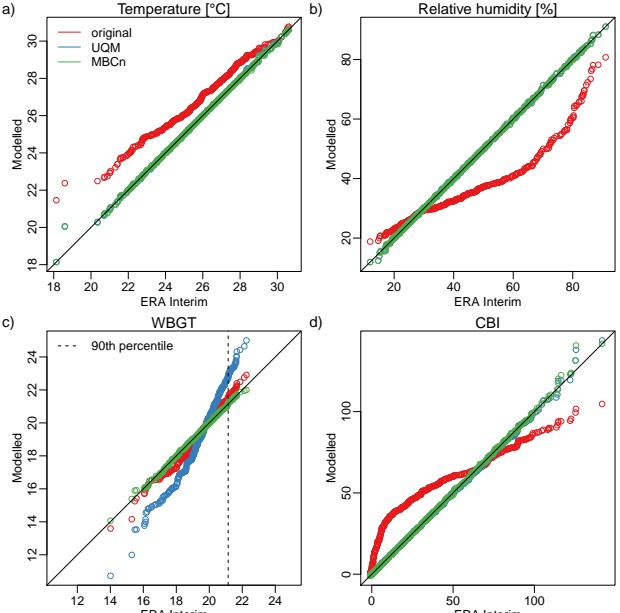

**Figure 2.** Quantile-quantile plot of observed versus modeled values for temperature (a), relative humidity (b), heat stress (WBGT, c) and fire risk (CBI, d) for a grid point in central Africa. Original model output (from IPSL-CM5A-MR) in red, model output corrected with UQM in blue, model output corrected with MBCn in green. The vertical dashed line in c) shows the 90th percentile of observed WBGT. The same data as in Fig. 1 were used. Note that in panels a), b) and d) the green circles cover most of the blue circles, as in these cases both bias adjustments yield virtually identical results.

Bias adjustment can also affect the relationship between a multivariate hazard indicator and its individual contributing

climate variables. For instance, looking at the raw model output of IPSL-CM5A-MR in Fig. 1b, one would infer that high heat stress (WBGT values exceeding their 95th percentile, highlighted in red) is reached at rather low relative humidity. Instead, for ERA-Interim data for the same grid point, high heat stress is associated with high relative humidity (Fig. 1a). UQM improves this mismatch to some extent but not entirely (Fig. 1c). Using MBCn for bias adjustment we would infer the correct contribution of individual variables. This is an important aspect, which needs to be taken into account when applying bias adjustment because it may lead to incorrect conclusions about which climate drivers are most relevant for extreme hazards and impacts. Consequently, we might also focus on the wrong aspects to improve in numerical climate models.

In the following we quantify whether the case illustrated in Figs. 1 and 2 is representative across the globe. We find that an increase in Root Mean Squared Error (RMSE) between the cumulative distribution functions of ERA Interim and model output before and after bias adjustment is an exception (Fig. 3a). However, even though biases decrease at most grid points when applying UQM, the reduction in bias is less than 50% at $15 \pm 6\%$ (mean $\pm$ one standard deviation across models) of all grid points (Fig. 3b). Reducing biases for extreme percentiles ($\geq$ q90) may be even more challenging. For instance, for the 90th (95th) percentile of WBGT, UQM applied on temperature and relative humidity results in WBGT estimates that have larger biases than the estimate based on raw model output for about $15 \pm 6\%$ ($18 \pm 8\%$) of all grid points (Fig. 3a). If we ask for a reduction in bias by at least 50%, $27 \pm 10\%$ ($32 \pm 12\%$) of grid points cannot reach this benchmark (Fig. 3b). Note that this means that in the majority of grid points, UQM reduces biases in WBGT. However, in many cases the reduction is not satisfactory. Understanding the conditions under which bias adjustment fails may help to design and use approaches that are more suitable for the given target. In contrast to WBGT, for CBI, UQM efficiently reduces the bias in most cases. Relative humidity alone explains most of the variance in CBI (Fig. 1), hence UQM can correct it very well (Fig. 2-3). If instead of UQM the bias adjustment used in ISIMIP2b is applied, the numbers mostly fall into the CMIP5 range for WBGT (Fig. 3). However, for CBI, more grid points show no substantial improvement after bias adjustment compared to CMIP5. This is probably due to the fact that UQM is extremely flexible and thus almost perfectly adjusts temperature and relative humidity. This in turn leads to a very good adjustment of CBI, whose variability largely follows relative humidity (Fig. 1), whereas the bias adjustment used in ISIMIP2b is more conservative and therefore less flexible in adjusting CBI.

Regions for which biases only slightly increase or decrease include locations where biases are small to start with. To study for how many grid points this is the case, we compute the fraction of grid points for which the bias in WBGT is larger than 1 Kelvin (K) either before or after bias adjust-

ment. This is the case for 50-90% of the grid points, depending on the model and the metric (Fig. 4a). Recomputing Fig. 3 based on this subset reduces the fraction of locations where bias adjustment does not achieve the two benchmarks by about half. Because these numbers strongly depend on the size of the accepted bias (1 K in our example), we continue the analysis with all grid points.

In Australia, the Sahel, some parts in sub-Saharan Africa and South America, UQM increases biases in the 90th percentile of WBGT for a large fraction of models (Fig. 5a). In those regions but also some other areas in the world, UQM reduces WBGT biases in the 90th percentile by less than 50% in the majority of all models (Fig. 5b). Overall, in nearly 11% of the land area, for more than 50% of the models, UQM does not reduce the bias of the 90th percentile of WBGT by more than 50%. MBCn on the other hand is able fit the hazard estimates well where UQM fails (not shown for all grid points but Fig. 1d and Fig. 2c illustrate the worst case).

To improve the usage of bias adjustment methods it would be important to know whether we can make any *a priori* statements as to whether UQM adjustment will lead to a reduction in biases of modeled multivariate hazards or impacts. Generally, regions where UQM fails to improve biases in WBGT are regions where the observed correlation between temperature and relative humidity is outside the CMIP5 range (Fig. 6a). UQM fails when (univariate) mean biases in temperature are small (Fig. 6b), and when the correlation between the driving variables (in this case temperature and relative humidity) is not captured well (Fig. 6d). The mean bias in relative humidity looks similar for both cases (Fig. 6c). Overall this means that if mean biases in climate drivers are large, any bias adjustment will lead to a substantial reduction in biases of hazard or impact indicators (consistent with Fig. 4a).

Internal variability can impact the effectiveness of bias adjustment because it introduces uncertainties that may be amplified by incomplete bias adjustment. For temperature and relative humidity individually, the perfect model approach reveals little difference between the range of the noise (uncertainty associated with internal variability) and the uncertainty range of all models (Fig. 7 illustrates the approach for the grid point of Austin, Texas, US). This is to be expected, as these driver variables can be perfectly adjusted. In the case of WBGT and CBI, however, UQM leads to much larger uncertainties for the full range than for the noise for some percentiles (Fig. 7). Overall, the full range of bias adjusted model simulations for the 90th percentile of WBGT is by a factor of 10 larger than the range expected from internal variability alone in many regions, including the Amazon, eastern North America and Indonesia (Fig. 8a). The regions with large increases in uncertainty roughly coincide with regions where the between-model variability in the correlation between temperature and relative humidity is very high compared to the variability within the CanESM runs (Fig. 8b). This is consistent with (Fig. 6d), which shows that univariate

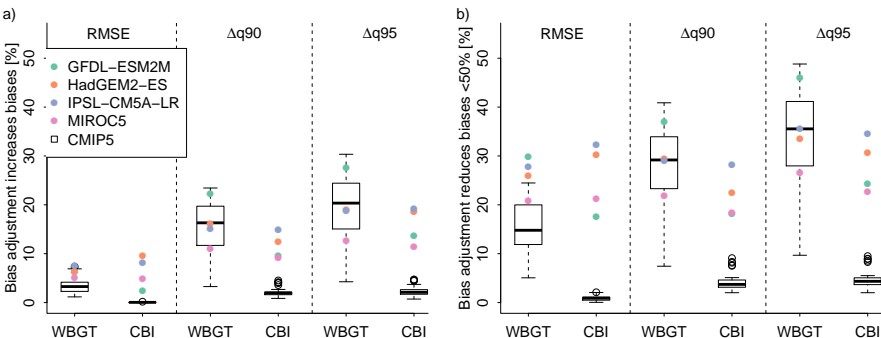

**Figure 3.** Fraction of grid points for which UQM increases bias (a) or does not result in a reduction of more than 50% in biases (b) of modeled hazards. Boxplots represent the CMIP5 multi-model ensemble after UQM and highlight the median (horizontal line), interquartile range (box), 1.5 times the interquartile range (whiskers), and outliers (points). Colored dots represent the results for the ISIMIP2b bias adjustment applied to the four GCMs used in ISIMIP2b. Shown are the metrics RMSE between empirical cumulative distribution functions and absolute differences in the 90th ($\Delta$q90) and 95th percentile ($\Delta$q95) between hazard indicators computed from (bias-adjusted) model output and observations (ERA-Interim for CMIP5 and EWEMBI for ISIMIP2b).

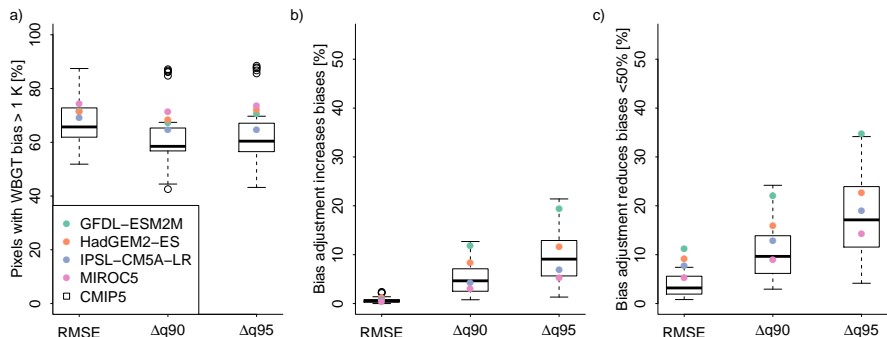

**Figure 4.** a) Fraction of grid points for which biases in WBGT are larger than 1 K before or after the application of bias adjustment. b-c) As in Figure 3 only for WBGT, based on the subset of grid points identified by a).

bias adjustment is not very successful when the correlation structure between models and reference is not matched well. For a fixed percentile threshold, the fraction of grid points for which the full range is at least two times the range of the noise varies between 10 and 40% for CBI, and between 30 and 70% for WBGT (Fig. 9). If we use MBCn to adjust biases, the uncertainty associated with internal variability is largely indistinguishable from the full uncertainty range (Fig. 9, dashed lines).

## 4   Discussion

We find that UQM of temperature and relative humidity often does not lead to a substantial reduction in WBGT biases. In a sizable number of cases UQM even leads to an increase in the original biases, particularly for high percentiles that are potentially most impact-relevant. The fire risk indicator CBI is less affected by these issues because, although temperature is required for calculating it, it is largely dominated by vari-

ations in relative humidity. Our findings on the limited effectiveness of univariate bias adjustment are admittedly based on one bias adjustment approach and two hazard indicators. Nevertheless, we expect that our results also hold for other multivariate hazards or impacts whose drivers are adjusted with bias adjustment approaches that do not explicitly correct for dependencies between variables.

We show that for nearly 40% of the land area, UQM fails to reduce biases in the 90th percentile of WBGT by at least 50% in more than half of the models. The challenges in correcting high percentiles of WBGT suggest that a direct application of UQM to a warmer climate may lead to large errors. Multivariate bias adjustment such as MBCn offer remedies, though for a generic impact modeling project such as ISIMIP, all variables and dependencies would need to be corrected at once, requiring large amounts of good climate observations to fill the high-dimensional data space (Cannon, 2018). Furthermore, observational datasets would need to be carefully tested as to whether they represent the desired dependencies

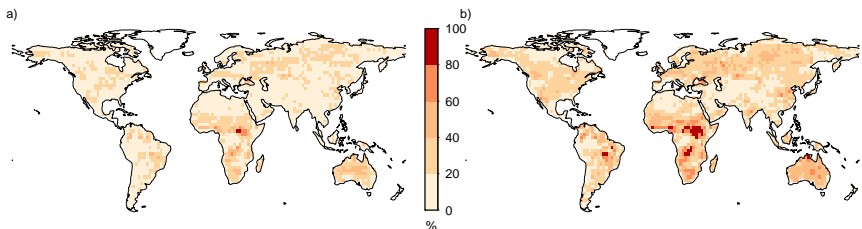

**Figure 5.** Fraction of models for which UQM increases biases (a) or does not decrease biases by more than 50% in the 90th percentile of WBGT (b).

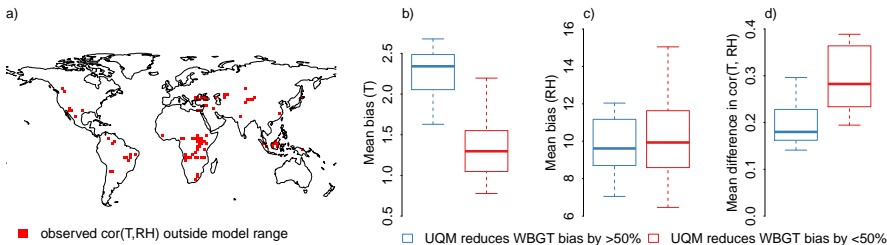

**Figure 6.** Reasons why UQM may fail. a) Regions in which the correlation between temperature and relative humidity in ERA Interim is outside the CMIP5 model range. b-d) Conditions where UQM does not lead to a substantial reduction in biases in WBGT (measured as difference in empirical cumulative distribution function RMSE between model output and ERA Interim before and after bias adjustment). Shown is the mean bias in temperature ($T$, b) and relative humidity ($RH$, c) as well as the mean difference in correlation over all grid cells between temperature and relative humidity (d). Blue (red) represents the distribution across models for cases in which UQM does (does not) lead to a reduction in biases of WBGT by at least 50%.

correctly (Cortés-Hernández et al., 2016; Zscheischler and Seneviratne, 2017).

The perfect model approach demonstrates that uncertainties related to internal variability and the use of an incomplete bias adjustment can lead to substantial uncertainty in multivariate hazards, particularly for more extreme percentiles. The large increase in uncertainty illustrated by the full range stems from a combination of initial uncertainty related to internal variability and the inability of UQM to adjust dependencies adequately. Therefore, uncertainties related to internal variability or other types of uncertainties may be strongly increased by incomplete bias adjustment. These uncertainties are likely sensitive to the length of the time period, as longer time periods will reduce the noise component related to internal variability due to a better sampling of the tails. These types of uncertainty are typically not communicated and accounted for in impact modeling chains. Hence, to be fully transparent, impact modelling should account for uncertainties associated with the chosen bias adjustment approach in addition to uncertainties related to the choice of climate model and impact model.

Our results challenge the general applicability of conclusions from previous studies that have investigated whether a bias adjustment that is separately applied to all components of a hazard indicator can effectively reduce biases in the latter. For instance, Yang et al. (2015) studied the effect of component-wise bias adjustment on a fire risk indicator in Sweden and conclude that bias adjusting more drivers reduces biases in the indicator without the need to consider the dependence between the drivers. Similarly, Casanueva et al. (2018) studied the implications of component-wise bias adjustment on fire risk in Spain and find that there is little difference between separately adjusting the drivers before computing the hazard indicator and adjusting the hazard estimate directly. Räty et al. (2018) conclude for hydrological variables that for most impacts, univariate methods have a comparable performance to bivariate methods. In contrast, Cannon (2018) clearly shows that including the dependence structure of drivers into the bias adjustment procedure strongly reduces biases in the Canadian Fire Weather Index. In general, we cannot draw the general conclusion that multivariate bias adjustment is not necessary in any case from individual, typically regional studies. Our findings suggest that whether bias adjustment approaches lead to a substantial improvements of impact indicators ultimately depends on (i) how large the initial model biases are, (ii) how strongly the indicator truly depends on multiple variables, and (iii) how well the models simulate relevant dependencies between the climate variables (Fig. 6). Overall, it is difficult to pin down under which exact circumstances univariate bias adjustment might fail. We assume that modeled impacts that truly depend on multiple dependent climate drivers are particularly susceptible to these issues. Impacts that fall into that category include heat related mortality (mostly depending on temperature and humidity

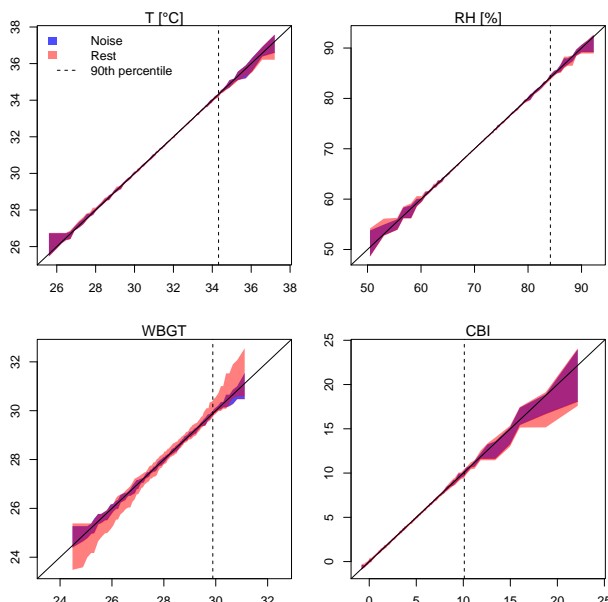

**Figure 7.** Illustration of the perfect model approach to study the effect of internal variability and UQM on the uncertainty range across models. Shown is the grid point of Austin, Texas, US. UQM was applied separately on temperature (T) and relative humidity (RH), using multiple runs (5) of CanESM as observations. CanESM runs bias adjusted against themselves represent the noise associated with internal variability (blue). The full range repressnts the range of all model simulations bias adjusted against all CanESM runs (red).

but also other factors), agricultural yields and carbon uptake (temperature, precipitation, radiation), drought (precipitation, evapotranspiration), and the security of energy supply in a system that combines input from various renewable energy sources (radiation, precipitation, wind speed, Sterl et al., 2018). Impacts that might be less affected by these issues are those that predominantly depend on a single climate variable such as runoff or floods, which mostly rely on precipitation (Räty et al., 2018). In these cases, the adjustment of the spatial and temporal distribution of precipitation might by more relevant than the adjustment of dependencies between precipitation and other climate variables. A bias adjustment method that is able to deal with very high dimensionality, for instance occurring when adjusting the covariance between many locations at the same time, was recently proposed by Vrac (2018).

While efforts to improve climate models and reduce their biases will continue (Wang et al., 2014; Davin et al., 2016; Kay et al., 2016), impact assessments need to become more transparent. In particular, uncertainties need to be well communicated to aid adaptation planning (Wilby and Dessai, 2010). Many artifacts of currently widely used bias adjustment methods are probably unknown because of the way bias adjustment is evaluated (Addor and Fischer, 2015; Maraun et al., 2017). Evaluation of multivariate relationships and ex-

tremes need to become standard in the evaluation of climate and impact models as well as an evaluation of the appropriateness of the chosen bias adjustment approach (Cortés-Hernández et al., 2016; Zscheischler et al., 2018). Overall, multivariate bias adjustment methods should be favoured in impact modeling to ensure that multivariate impacts are captured more realistically.

As long as major biases in climate models persists, some form of bias adjustment is unavoidable to model climate impacts. In the fortunate case where large multimodel ensembles are available and a clear target is identified, ensemble members can be selected according to how well they match certain criteria associated with the target (Sippel et al., 2016; Maraun et al., 2017; Herger et al., 2018). Process-based observational constraints (Hall and Qu, 2006; Sippel et al., 2017; Vogel et al., 2018) are one way forward to select the most promising subset of models (Maraun et al., 2017). Yet in many cases, this information is not available and large model ensembles may be too expensive to obtain. To achieve the most reliable outcome with respect to modeled impacts, bias adjustment preferably takes into account the known limitations of the relevant climate model as well as characteristics of the target system for which the bias adjustment is applied (Maraun et al., 2017). In large-scale impact modeling projects such as ISIMIP, global flood modeling (Winsemius et al., 2015), global crop modeling (Ruane et al., 2017), as well as generic modeling of high-impact events (Done et al., 2015), achieving these standards is extremely challenging, if not unfeasible. In these cases, typically, a single bias adjustment method is applied to a set of climate variables that then serves as input for a variety of impact models across regions and sectors. The usefulness of such one-size-fits-all approaches may be debated (Maraun et al., 2017) yet decision makers urgently require robust information on potential impacts of climate change. Impact modeling framework such as ISIMIP and many others sample the uncertainty related to the chosen climate model and impact model but do not take into account uncertainties related to the applied bias adjustment method. To be transparent towards potential users, however, the scientific community should provide information on all uncertainties associated with modeled impacts.

## 5   Conclusions

Climate impact modeling is crucial to translate information from climate projections into potential impacts to aid decision making and planning. Due to persistent biases in current climate models, bias adjustment is an integral part of most impact modeling activities. Our results demonstrate that univariate bias adjustment can increase biases of simulated hazards that depend on multiple correlated climatic drivers. Univariate bias adjustment can furthermore lead to a large increase in uncertainty of such modeled hazards and impacts. Both aspects are particularly severe when studying extremes.

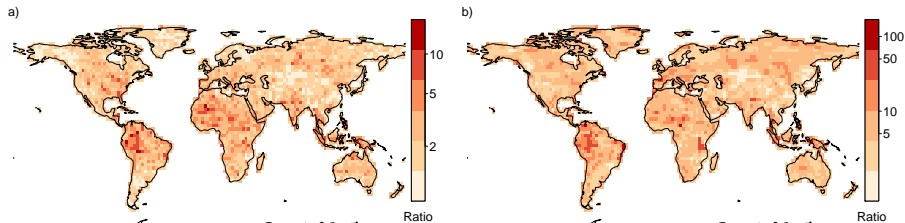

**Figure 8.** (a) Ratio between full range and noise (uncertainty associated with internal variability) at the 90th percentile of heat stress (WBGT) based on a perfect model approach using UQM to adjust biases. (b) Ratio between the range of all models and the range of all 5 CanESM runs of correlation between temperature and relative humidity.

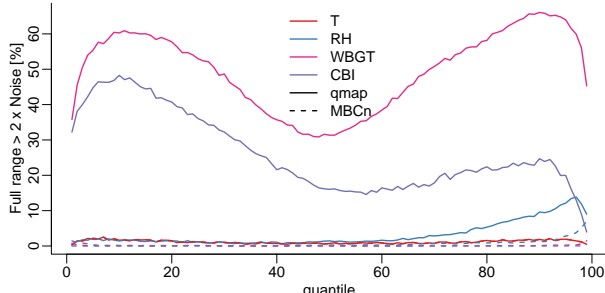

**Figure 9.** Fraction of grid points for which the full range is at least twice the range of the noise (uncertainty associated with internal variability) in the perfect model approach. Solid lines represent cases computed with UQM, dashed lines represent cases computed with MBCn.

More importantly, if univariate bias adjustment does not adequately adjust biases in hazards in present-day climate, future projections of such hazards have to be interpreted very carefully. Our findings highlight that impact modeling chains need to incorporate uncertainties associated with the choice of bias adjustment into their uncertainty assessment to be transparent for decision makers.

**Code and data availability.** All datasets and the code to compute the different bias adjustments are freely available from the sources mentioned in the Section 2 (Data and Methods).

**Appendix A**

**Table A1.** The 29 CMIP5 models used in this study. We use daily data from the historical simulation over 1981-1995.

| Model name | Modeling center | Initialization |
|---|---|---|
| ACCESS1.0 | Commonwealth Scientific and Industrial Research Organization (CSIRO) and Bureau of Meteorology (BOM), Australia | r1i1p1 |
| ACCESS1.3 | Commonwealth Scientific and Industrial Research Organization (CSIRO) and Bureau of Meteorology (BOM), Australia | r1i1p1 |
| BCC-CSM1.1 | Beijing Climate Center, China Meteorological Administration | r1i1p1 |
| BCC-CSM1.1M | Beijing Climate Center, China Meteorological Administration | r1i1p1 |
| CanESM2 | Canadian Centre for Climate Modelling and Analysis | r1i1p1-r1i1p5 |
| CNRM-CM5 | Centre National de Recherches Météorologiques / Centre Européen de Recherche et Formation Avancée en Calcul Scientifique | r1i1p1 |
| CSIRO-Mk3.6.0 | Commonwealth Scientific and Industrial Research Organization in collaboration with Queensland Climate Change Centre of Excellence | r1i1p1 |
| GFDL-CM3 | NOAA Geophysical Fluid Dynamics Laboratory | r1i1p1 |
| GFDL-ESM2G | NOAA Geophysical Fluid Dynamics Laboratory | r1i1p1 |
| GFDL-ESM2M | NOAA Geophysical Fluid Dynamics Laboratory | r1i1p1 |
| INM-CM4 | Institute for Numerical Mathematics | r1i1p1 |
| IPSL-CM5A-LR | Institut Pierre-Simon Laplace | r1i1p1-r1i1p4 |
| IPSL-CM5A-MR | Institut Pierre-Simon Laplace | r1i1p1 |
| IPSL-CM5B-LR | Institut Pierre-Simon Laplace | r1i1p1 |
| MIROC-ESM | Japan Agency for Marine-Earth Science and Technology, Atmosphere and Ocean Research Institute (The University of Tokyo), and National Institute for Environmental Studies | r1i1p1 |
| MIROC-ESM-CHEM | Japan Agency for Marine-Earth Science and Technology, Atmosphere and Ocean Research Institute (The University of Tokyo), and National Institute for Environmental Studies | r1i1p1 |
| MIROC5 | Atmosphere and Ocean Research Institute (The University of Tokyo), National Institute for Environmental Studies, and Japan Agency for Marine-Earth Science and Technology | r1i1p1-r1i1p3 |
| MRI-CGCM3 | Meteorological Research Institute | r1i1p1 |
| MRI-ESM1 | Meteorological Research Institute | r1i1p1 |
| NorESM1-M | Norwegian Climate Centre | r1i1p1 |

**Author contributions.** J.Z. and E.M.F. conceived the study. S.L. computed the bias adjustment following the ISIMIP2b scheme. J.Z. analysed all data and produced all figures. J.Z. wrote the first draft, all authors commented on the draft and all revisions.

**Competing interests.** The authors declare that they have no conflict of interest.

**Acknowledgements.** We thank Alex Cannon for helpful discussions related to the application of the MBCn approach. Jakob Zscheischler acknowledges financial support from the SNSF (Ambizione grant PZ00P2_179876). Stefan Lange acknowledges funding from the European Union's Horizon 2020 research and innovation programme under grant agreement No 641816 (CRESCENDO).

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
