# Peer review of "The effect of univariate bias adjustment on multivariate hazard estimates"

_Earth System Dynamics, 2018_

## Referee Comment (RC1) · Anonymous Referee #1 · 8 Oct 2018

This manuscript provides a valuable contribution to the literature on bias correction, focusing on the issue of 'handling' inter-variable dependencies and consequences for derived impact metrics (here a heat stress index and a fire index). Overall the paper is well written and figures are of good quality. My comments are minor in nature and often addressed by technical edits.

My comments are as follows: 1. It would be meaningful to see an argument for why you are focusing on a global scale here (using GCM output) rather than output from regional climate models, that typically provide outputs used for impact models. I can see motivations for this, e.g. spatial (global) completeness, addressing the source of the change signal (as provided by the GCM – and then translated to a finer resolution by a RCM). I have no objection to the GCM focus but given that bias correction is typically

a problem for impact studies, and many of these use downscaled data, you might want to provide a motivation for the experimental setup. I think it is also noteworthy that in the context of downscaling, some argue for bias correcting the input fields to the RCM –so to avoid propagation of error in the RCM. You might also want to talk to/refer to the issues of dealing with spatial dependencies – if corrections are applied to grid cells, how is spatial dependencies (and indeed temporal dependencies) preserved/modified. 2. I think you need a more detailed description of the model simulation datasets used in this study. I don't think it is enough to list what projects they are associated with, it would be meaningful to have details such as ensemble configuration, range of model resolutions, use of initial condition members (or not) etc. Under 'data' you could provide details on the re-analysis dataset as well as on model ensembles (if you wanted to keep obs from model simulations you could use different sub headings). As it currently reads, different model names crop up in various places of the text and figures, which causes a bit of confusion upon reading. 3. I think it would make sense to explicitly state (in an appropriate place in the introduction) that we assume that bias correction to stationary, i.e. that it is valid to develop a correction under current climate and apply this in a warmer world. 4. I would consider putting the bias correction methods into the manuscript as mathematical formula – this would enhance clarity in terms of understanding the methods, and it makes the paper self-contained (rather than pointing to another paper for understanding the specifics of the method). 5. In section 15, instead of writing 'We then' I wonder if you should start with 'firstly' (or similar) – to reflect that this is the first step of the analysis? Or perhaps I have misunderstood. 6. In the same section as above, I think it would be helpful to be more specific about what you mean with 'all other runs', all other CMIP5 current climate runs – all of them? Also, do you bias correct towards all of the CanESM ensemble members - all five? I wonder for this type of paper if you might want to think about some form of infographics, illustrating your experiment setup, what comparisons are made etc.

---

## Referee Comment (RC2) · S. Hagemann (Referee) · 19 Oct 2018

**Manuscript:** The effect of bias adjustment on impact modeling

**Major remarks**

The authors present an interesting study on how univariate bias corrections of climate model output affect impact indicators that depend on more than one climate variable. They chose two hazard indicators related to heat stress and fire risk to demonstrate to demonstrate the effect of separate univariate bias corrections in comparison with a multivariate method. The latter corrects the dependence structure between the variables in addition to the respective variable distributions. The paper is written well so that I have only a few minor comments.

- The title seems to be too general. The focus of the paper is on impact indicators that depend on more than one climate variable, and not on impact modelling in general. I suggest revising the title.

- I suggest citing (in the introduction and discussion of results) Räty et al. (2018) who actually found that in many cases a multivariate bias correction is not necessary (from the hydrological perspective). They stated that "the additional benefit of using bi-variate bias correction methods is not obvious, as univariate methods have a comparable performance. "
  Räty, O.; Räisänen, J.; Bosshard, T.; Donnelly, C. Intercomparison of Univariate and Joint Bias Correction Methods in Changing Climate From a Hydrological Perspective. Climate 2018, 6, 33.

- The analysis of results (Sect. 3) takes into account relative changes in the bias (reduction by at least 50%, increase). This means that also grid points are included where the bias is small/negligible for impact purposes. Here, a low reduction in bias or even a small increase in the bias would not matter for modelling the impacts. Is there a way of setting a bias threshold that defines the 'acceptable' bias, and then consider only the noteworthy changes on biases above this threshold? This means to include only points in the analysis where the bias before or after correction is above this threshold. I think that such a discrimination is helpful to judge how problematic the application of univariate bias correction is for those biases that matter. One results of the study is that univariate bias correction cannot effectively reduce biases in multivariate hazard estimates when (iii) univariate biases are small. However, if the resulting biases in the hazard indicator are small, this will not matter for the respective cases.

I suggest accepting the paper for publication after minor revisions are conducted.

**Minor remarks**

In the following suggestions for editorial corrections are marked in *Italic*.

Fig. 3
I suggest adding one line that indicate the type of each column for which WBGT and CBI are considered, i.e. RMSE, Δq90 and Δq95.

Fig. 4 and 7
I suggest using a discrete colour bar to improve the respective figures.

Fig 5.
It is difficult to identify regions in panel a). I suggest using another colour to indicate the regions, e.g. red.

p.10 – line 21
… period, *as longer* time …

p.12 – line 23
*We* thank Alex …

p.17 – line 20
… explain *a large* fraction …

---

## Author Comment (AC1) · 9 Nov 2018

*We thank the reviewer for the helpful comments. Our replies below are highlighted in italic.*

Reviewer 1:
This manuscript provides a valuable contribution to the literature on bias correction, focusing on the issue of 'handling' inter-variable dependencies and consequences for derived impact metrics (here a heat stress index and a fire index). Overall the paper is well written and figures are of good quality. My comments are minor in nature and often addressed by technical edits.

My comments are as follows:
1. It would be meaningful to see an argument for why you are focusing on a global scale here (using GCM output) rather than output from regional climate models, that typically provide outputs used for impact models. I can see motivations for this, e.g. spatial (global) completeness, addressing the source of the change signal (as provided by the GCM – and then translated to a finer resolution by a RCM). I have no objection to the GCM focus but given that bias correction is typically a problem for impact studies, and many of these use downscaled data, you might want to provide a motivation for the experimental setup. I think it is also noteworthy that in the context of downscaling, some argue for bias correcting the input fields to the RCM –so to avoid propagation of error in the RCM. You might also want to talk to/refer to the issues of dealing with spatial dependencies – if corrections are applied to grid cells, how is spatial dependencies (and indeed temporal dependencies) preserved/modified.

*Our focus on GCMs is motivated by global-scale impact modelling frameworks (assessing flood risk, crop impacts etc., as e.g. performed within ISIMIP). ISIMIP for instance remaps coarse-scale GCMs to 0.5 degrees and then applies bias adjustment. It is correct that for more local assessments often RCMs are used. However, since our study is quite general and of a more conceptual nature, the issues raised here also apply to RCMs. We will slightly revise the introduction and motivate the use of GCM output better.*

*Adjusting spatial and temporal dependencies might indeed be relevant for a number of impacts. Yet, selecting the appropriate spatial and temporal scales for each location and adjusting time, space and multiple variables at the same time seems rather infeasible at the global scale as it would lead to an explosion in the number of dimensions that need to be bias adjusted. We mention this aspect for hydrological impacts on P11 L21: "In these cases, the adjustment of the spatial and temporal distribution of precipitation might by more relevant than the adjustment of dependencies between precipitation and other climate variables." We agree however that this an important point to comment on and will extend the discussion in this regard. In particular, we will refer to a recent paper by M Vrac (Multivariate bias adjustment of high-dimensional climate simulations: the Rank Resampling for Distributions and Dependences ($R^2D^2$) bias correction, HESS, 22, 3175-3196, 2018) which proposes a bias adjustment method that can adjust spatial dependencies and works for a very high dimensionality.*

2. I think you need a more detailed description of the model simulation datasets used in this study. I don't think it is enough to list what projects they are associated with, it would be meaningful to have details such as ensemble configuration, range of model resolutions, use of initial condition members (or not) etc. Under 'data' you could provide details on the reanalysis dataset as well as on model ensembles (if you wanted to keep obs from model simulations you could use different sub headings). As it currently reads, different model names crop up in various places of the text and figures, which causes a bit of confusion upon reading.

*Thank you for this comment. All models are taken from the CMIP5 archive, probably one of the most used model archives in current climate research. Listing the model configurations of all used models here (nearly 30) is not very informative for the readers as it has no relevance for the results. We therefore refer here to the original publication (Taylor et al., 2012). Nevertheless, we agree that in its current form the data section is not so easily accessible to researchers who are not familiar with the used datasets. We therefore extend the description of the used datasets and provide information on the rationale of the model simulations and their usage as well as more detailed information on the observational datasets.*

3. I think it would make sense to explicitly state (in an appropriate place in the introduction) that we assume that bias correction to stationary, i.e. that it is valid to develop a correction under current climate and apply this in a warmer world.

*Thank you, we will add this information to the introduction.*

4. I would consider putting the bias correction methods into the manuscript as mathematical formula – this would enhance clarity in terms of understanding the methods, and it makes the paper self-contained (rather than pointing to another paper for understanding the specifics of the method).

*Thank you for this suggestion. We will add more information, including formulas, on the used bias correction methods to make the paper more self-contained.*

5. In section 15, instead of writing 'We then' I wonder if you should start with 'firstly' (or similar) – to reflect that this is the first step of the analysis? Or perhaps I have misunderstood.

*We agree and will adjust the wording.*

6. In the same section as above, I think it would be helpful to be more specific about what you mean with 'all other runs', all other CMIP5 current climate runs – all of them? Also, do you bias correct towards all of the CanESM ensemble members - all five? I wonder for this type of paper if you might want to think about some form of infographics, illustrating your experiment setup, what comparisons are made etc.

*We mean all other runs of the CMIP5 subset used in this study (i.e., from 29 model simulations of the historical time period). We bias correct towards all CanESM runs to propagate the uncertainty related to internal variability through the bias adjustment. We will think about a small infographic to illustrate the perfect model approach.*

---

## Author Comment (AC2) · 9 Nov 2018

*We thank the Stefan Hagemann for his helpful comments. Our replies below are highlighted in italic.*

The authors present an interesting study on how univariate bias corrections of climate model output affect impact indicators that depend on more than one climate variable. They chose two hazard indicators related to heat stress and fire risk to demonstrate to demonstrate the effect of separate univariate bias corrections in comparison with a multivariate method. The latter corrects the dependence structure between the variables in addition to the respective variable distributions. The paper is written well so that I have only a few minor comments.

- The title seems to be too general. The focus of the paper is on impact indicators that depend on more than one climate variable, and not on impact modelling in general. I suggest revising the title.

*We agree and will change the title to "The effect of univariate bias adjustment on multivariate hazard estimates".*

- I suggest citing (in the introduction and discussion of results) Räty et al. (2018) who actually found that in many cases a multivariate bias correction is not necessary (from the hydrological perspective). They stated that "the additional benefit of using bivariate bias correction methods is not obvious, as univariate methods have a comparable performance. "

Räty, O.; Räisänen, J.; Bosshard, T.; Donnelly, C. Intercomparison of Univariate and Joint Bias Correction Methods in Changing Climate From a Hydrological Perspective. Climate 2018, 6, 33.

*Thank you for this suggestion. Indeed, we already mention that in the hydrological context, the discussed issues are likely not very important because often precipitation is the dominant driver (p. 11 l. 19). However, even though many hydrological impacts might not be affected by incorrect multivariable dependencies, this might look different for impacts that strongly depend on multiple variables. Based on individual examples where multivariate bias adjustment did not lead to improvements, we cannot draw the general conclusion that multivariate bias adjustment is not necessary in any case. This is discussed in on p. 11 l. 4 onwards. We will include the paper by Räty et al in this discussion.*

*Räty et al. also point out that differences between impact simulations based on climate input data bias-adjusted with univariate versus multivariate methods are most often smaller in cross-validation than in validation metrics. This also does not imply that multivariate bias adjustment is generally unnecessary, as cross-validation can results can be strongly misleading (Maraun & Widmann, 2018). We will extend our discussion section towards this point and refer to Räty et al. in that context.*

- The analysis of results (Sect. 3) takes into account relative changes in the bias (reduction by at least 50%, increase). This means that also grid points are included where the bias is small/negligible for impact purposes. Here, a low reduction in bias or even a small increase in the bias would not matter for modelling the impacts. Is there a way of setting a bias threshold that defines the 'acceptable' bias, and then consider only the noteworthy changes

on biases above this threshold? This means to include only points in the analysis where the bias before or after correction is above this threshold. I think that such a discrimination is helpful to judge how problematic the application of univariate bias correction is for those biases that matter. One results of the study is that univariate bias correction cannot effectively reduce biases in multivariate hazard estimates when (iii) univariate biases are small. However, if the resulting biases in the hazard indicator are small, this will not matter for the respective cases.

*We agree that in many cases the absolute bias might be small. In general it is often difficult to choose the magnitude of an acceptable bias and this will be highly context- and variable dependent. Nevertheless, we agree that this is an important point to discuss. Hence, we compute the fraction of pixels for which the bias in WBGT is larger than 1 either before or after bias adjustment. This is the case for 50-90% of the pixels, depending on the model and the metric. We then recompute figure 3 based on this subset (figure shown below). The resulting fraction of locations where bias adjustment does not achieve the two chosen benchmarks is about half of the original numbers when all locations are included. We will include this figure in the revision and discuss its implications. However, because these numbers strongly depend on the size of the accepted bias, we will keep the other figures as is.*

[Figure]

*Figure: a) Fraction of pixels for which biases in WBGT are larger than 1 K before or after the application of bias adjustment. b-c) As in Figure 3 only for WBGT, based on the subset of pixels identified by a).*

I suggest accepting the paper for publication after minor revisions are conducted.

Minor remarks
In the following suggestions for editorial corrections are marked in Italic.

Fig. 3
I suggest adding one line that indicate the type of each column for which WBGT and CBI are considered, i.e. RMSE, \Delta q90 and \Delta q95.

Fig. 4 and 7
I suggest using a discrete colour bar to improve the respective figures.

Fig 5.

It is difficult to identify regions in panel a). I suggest using another colour to indicate the regions, e.g. red.

p.10 – line 21

… period, *as longer* time …

p.12 – line 23

*We* thank Alex …

p.17 – line 20

… explain *a large* fraction …

*Thank you. We will include all suggestions in the revised version.*